

# Residual block fully connected DCNN with categorical generalized focal dice loss and its application to Alzheimer's disease severity detection

Adi Alhudhaif[1] and Kemal Polat[2]

[1] Department of Computer Science, College of Computer Engineering and Sciences in Al-kharj, Prince Sattam Bin Abdulaziz University, Al-kharj, Saudi Arabia
[2] Department of Electrical and Electronics Engineering, Faculty of Engineering, Bolu Abant Izzet Baysal University, Bolu, Turkey

## ABSTRACT

**Background**. Alzheimer's disease (AD) is a disease that manifests itself with a deterioration in all mental activities, daily activities, and behaviors, especially memory, due to the constantly increasing damage to some parts of the brain as people age. Detecting AD at an early stage is a significant challenge. Various diagnostic devices are used to diagnose AD. Magnetic Resonance Images (MRI) devices are widely used to analyze and classify the stages of AD. However, the time-consuming process of recording the affected areas of the brain in the images obtained from these devices is another challenge. Therefore, conventional techniques cannot detect the early stage of AD.

**Methods**. In this study, we proposed a deep learning model supported by a fusion loss model that includes fully connected layers and residual blocks to solve the above-mentioned challenges. The proposed model has been trained and tested on the publicly available T1-weighted MRI-based KAGGLE dataset. Data augmentation techniques were used after various preliminary operations were applied to the data set.

**Results**. The proposed model effectively classified four AD classes in the KAGGLE dataset. The proposed model reached the test accuracy of 0.973 in binary classification and 0.982 in multi-class classification thanks to experimental studies and provided a superior classification performance than other studies in the literature. The proposed method can be used online to detect AD and has the feature of a system that will help doctors in the decision-making process.

## INTRODUCTION

The brain is a vital organ that contains memory and manages thoughts and decision-making (*Armstrong et al., 2009*). Alzheimer's disease (AD) is one of the causes of dementia. In AD, beta-amyloid and phosphorylated tau proteins accumulate excessively, which causes brain cell degeneration (*Anonymous, 2022*). AD is an irreversible disease that causes progressive brain deterioration. Memory cells gradually die, causing an increasing shrinkage in the

Corresponding author
Adi Alhudhaif,
a.alhudhaif@psau.edu.sa

brain (*Alberdi et al., 2018*). AD is a fatal neurological disease with a life expectancy of 4–8 years after diagnosis (*Babalola et al., 2009*). This disease, which can be seen at any age, is more common in people over 65 (*Coppola et al., 2013*; *Chandra, Dervenoulas & Politis, 2019*).

According to official figures in the United States, 121,499 people died from AD in 2019. AD was ranked sixth among the causes of death in 2019 when COVID-19 was among the top 10 reasons. In addition, In 2020 and 2021, it was shown as the seventh leading cause of death. Total payments made in 2022 for health care, long-term care, and hospice services for people aged 65 and over with dementia are estimated at $321 billion (*Anonymous, 2022*).

Several different computer-assisted neuroimaging methods can diagnose AD. In light of clinical experience, MRI has become almost standardized in these imaging modalities. Although AD is incurable, its progression can be slowed with early diagnosis and treatment (*Gopinadhan, Angeline Prasanna & Anbarasu, 2022*).

In the last decades, many machine learning-based and specifically deep learning-based approaches to diagnosing AD have been presented. If we summarize the previous studies in the literature, classification based on segmentation of brain images was used for early diagnosis of AD by *Mehmood et al. (2021)* For the AD classification, the VGG network was used for transfer learning. *Lahmiri (2023)* used deep convolutional neural networks (DCNN) to increase the diagnostic sensitivity of AD. Besides, the KNN approach was used to filter the number of features. *Shanmugam et al. (2022)* analyzed the success of three pre-trained networks, GoogLeNet, AlexNet, and ResNet-18, in classifying AD's stages. *Frizzell et al. (2022)* conducted a systematic literature review covering studies conducted between 2009 and 2020 for the diagnosis of AI-based AD using the PubMed database. In the review, images consisting of 3 different classes, including normal aging, mild cognitive impairment, and AD, obtained from MRI, were examined and comparatively analyzed with the proposed artificial intelligence algorithms. *Sathish Kumar et al. (2022)* have proposed a classification model that uses the AlexNet framework to diagnose AD at an early stage from MR images. *Jung, Luna & Park (2023)* proposed a new conditional generative adversarial network (cGAN) capable of synthesizing high-quality 3D MR images. The proposed model consists of an attention-based 2D generator, a 2D parser, and a 3D splitter that can synthesize 2D slices from 3D MR images. *Liu et al. (2023)* proposed the Monte Carlo aggregated neural network model combining ResNet50 and Monte Carlo sampling for early AD detection. The proposed model is trained on 2D slices obtained from 3D MR images. *Sharma et al. (2022)* proposed an artificial neural network model using the VGG16 deep learning network as a feature extractor for the classification of four different stages of AD. *Alorf & Khan (2022)* proposed a multi-label spoofing model using the Stacked Sparse Autoencoder and Brain Connectivity Graph convolutional network models for the six stages of AD obtained from the rs-fMR imaging device. *Raghavaiah & Varadarajan (2021)* trained a DCNN-based model on fMRI and MRI to diagnose AD from specific sound control information. It is fed to the proposed model *via* an image converter with decomposed parameters from fMRI and MRI. *Loddo, Buttau & Di Ruberto (2022)* proposed a deep learning model combining AlexNet, ResNet101, and Inception-ResNet-V2 for AD

classification. Studies in the literature achieved test success of up to 99% in ADNI and OASIS data sets with deep learning models. As far as we know in the literature, only *Loddo, Buttau & Di Ruberto (2022)* and *Sharma et al. (2022)* tested their models on the Kaggle dataset. The Kaggle dataset is a dataset that challenges models due to both its small size and difficulties in distinguishing between classes.

Manual classification of AD on brain images obtained from MRI devices is time-consuming. At the same time, AD is very similar to what happens in the brain with aging. Therefore, it is a challenging task to diagnose AD by clinicians. Thus, deep learning-based computer-aided systems, which often achieve higher success than clinicians, gain more significant importance for AD classification.

In summary, the main contributions of this paper to the literature are given as follows:

- In this study, we divided an inspired by the encoder layer of the U-shaped segmentation algorithm and connected the model we designed to the fully connected layer. Therefore, we have fused the Generalized Dice Loss (GDL) function used in multi-class segmentation with the Focal Loss (FL) function in such a network.
- Non-Local Means and Estimate sigma algorithms are first fused to eliminate noise in MR images by us.
- Contrast Limited Adaptive Histogram Equalization (CLAHE) algorithm used for histogram equalization to enhance MR images.
- A fully connected deep convolutional neural network with residual blocks is proposed to classify AD.
- A fusion loss function increased the network's test accuracy in classifying AD from MR images.

The second section, consisting of the material and method section, gives detailed information about the pre-processing stages of the data set and the proposed model. The practical applications of the model and the test results obtained from the proposed model are evaluated and discussed in the discussion section. Finally, the work done in the Conclusions section is summarized, and suggestions for future studies are shared.

## MATERIALS & METHODS

### MRI dataset

The KAGGLE online community obtained the dataset (*Dubey, 2023*). The dataset obtained from KAGGLE consists of 5,121 axial images collected from different websites. The images in the dataset were collected and labeled into four different classes. These are no dementia, very mild dementia, mild dementia, and moderate dementia. There is no information about the age of the patients in the MR images obtained from the patients. The training dataset consists of MR images of 2,560 healthy (ND), 1,792 very mild dementia (vmD), 717 mild dementia (miD), and 52 moderate dementia (mD) individuals. The test dataset consists of MR images of 640 healthy (NC), 448 very mild dementia (vmD), 179 mild dementia (miD), and 12 moderate dementia (mD) individuals. The resolution of the images is 176 × 208. Samples are shown in Fig. 1. No other circumstances of the MR images obtained were specified.

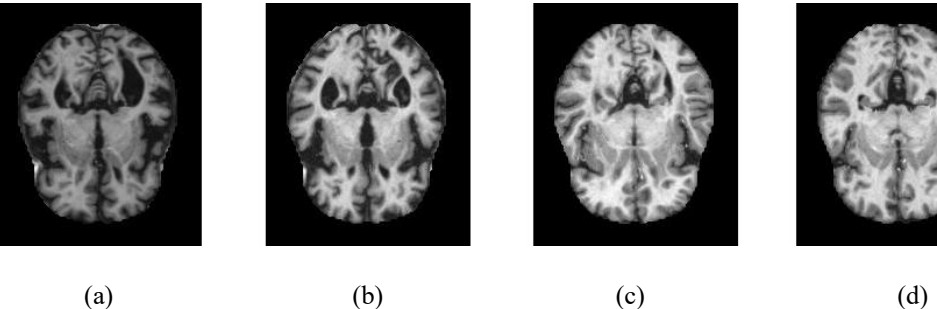

(a)          (b)          (c)          (d)

**Figure 1** **Samples of axial images in the Kaggle data set.** (A) No dementia, (B) very mild dementia, (C) mild dementia, (D) moderate dementia.

## Dataset pre-processing

Various image restoration techniques are used in the Kaggle AD dataset to increase the model's training and test accuracy. First, the Non-Local Means (NLM) algorithm was applied to the images. Then, the image noise was de-noised using the weighted average of the pixel neighborhoods with similarity. Also, to improve performance when calculating the similarity of each pixel, instead of considering just one pixel, a small area of pixels is chosen around it, given by the small_window parameter. Patch_Size = 2 and Patch_Distance = 1 were selected to increase the noise removal performance of the NLM algorithm. Finally, Estimate_sigma is fused with the NLM algorithm to improve the performance of the proposed model.

The resulting images were passed through the CLAHE algorithm. Contrast-limited histogram equalization is performed by dividing images into small blocks called tiles in CLAHE in the OpenCV library. As a result of the experimental studies, the clipping level in CLAHE was chosen as 2.6. Samples of four different classes are shown in Fig. 2, showing the MR images in the Kaggle dataset before and after the noise removal and CLAHE. In Fig. 2, the pictures in the top line are the MR images before the image pre-processing techniques, and the images in the bottom line are the MR images obtained after the image pre-processing methods. The pre-processed training data set was doubled by applying random shifting, random rotation, random rescaling, and random horizontal and vertical flip methods.

## The proposed categorical generalized focal dice loss function

The proposed Hybrid Loss function is a fused model of the Generalized Dice Loss (GDL) and Focal Loss (FL) (*Sudre et al., 2017*; *Lin et al., 2020*). The FL function formula is shown in Eq. (1).

$$FL = -\alpha_t (1 - p_t)^\gamma \log(p_t) \tag{1}$$

As seen in Eq. (1), contrary to Cross-entropy (CE), Loss of Focus prevents suppression of CE Loss in large class imbalances. Thus, the negatives that make up most of the loss are reduced. Adding $\alpha$ to Focal Loss in Balanced Cross Entropy balances positive and negative

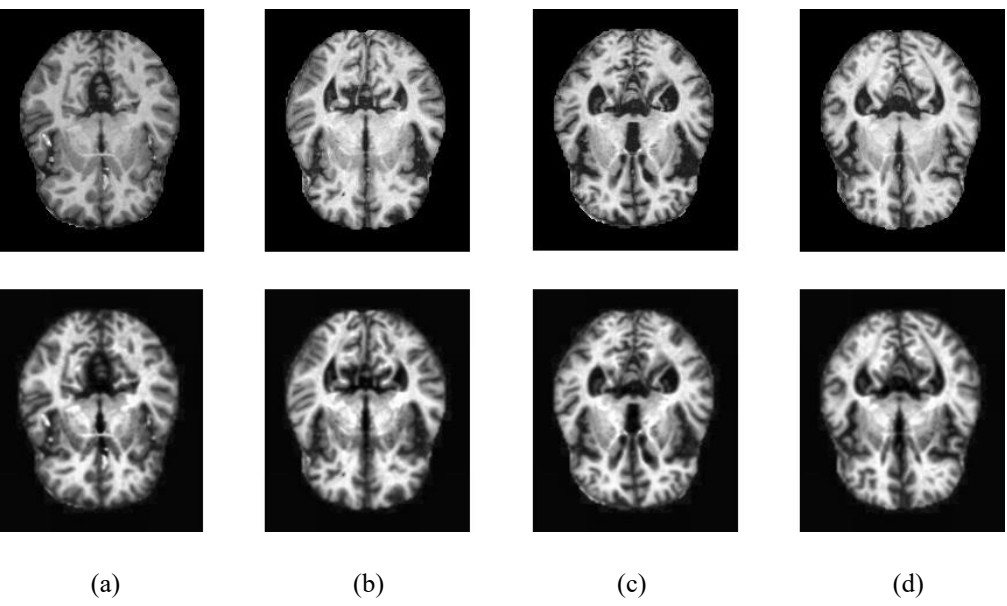

|          |          |          |          |
|:--------:|:--------:|:--------:|:--------:|
| (a) | (b) | (c) | (d) |

**Figure 2** **Samples of image pre-processing techniques in the Kaggle data set.** (A) No dementia, (B) very mild dementia, (C) mild dementia, (D) moderate dementia.

samples. In addition, the adjustable focusing parameter $\gamma \geq 0$ and the cross-entropy loss modulating factor $(1 - p_t)$ was added to reduce the weight of easy-to-learn samples and to enable the deep-learning model to focus on difficult-to-train samples. As a result of the experimental studies performed in this study, we chose $\gamma = 2$ and $\alpha = 4$, where $p$ is the model's predicted probability for the class (*Lin et al., 2020*).

The most significant disadvantage of Dice Loss is that although it has a good training score, its test score is low due to its poor response to class imbalances. As a solution to this, *Sudre et al. (2017)* transformed the Generalized Dice Score (GDS) function proposed by *Crum, Camara & Hill (2006)* to score multi-class segmentation into a loss function named Generalized Dice Loss (GDL). The GDL function formula is shown in Eq. (2).

$$GDL = 1 - 2 \frac{\sum_{l=1}^{2} w_l \sum_n r_{ln} p_{ln} + \in}{\sum_{l=1}^{2} w_l \sum_n r_{ln} + p_{ln} + \in} \tag{2}$$

where $r_l$ is the ground truth, and $p_l$ is the predicted value. $\in$ is a smoothing value for fine-tuning.

Here $w_l$ isused to provide immutability to different labeled sets where $w_l$ is formulated as shown in Eq. (3).

$$w_l = \frac{1}{\left(\sum_{n=1} r_{ln}\right)^2} \tag{3}$$

where $r_l$ is the ground truth. In this study, FL and GD, seen as robust models in multi-class segmentation, were fused to predict four different classes of AD, as shown in Eq. (4).

$$Fusion\_Loss = (1 - lamda) * GDL + lamda * FL \tag{4}$$

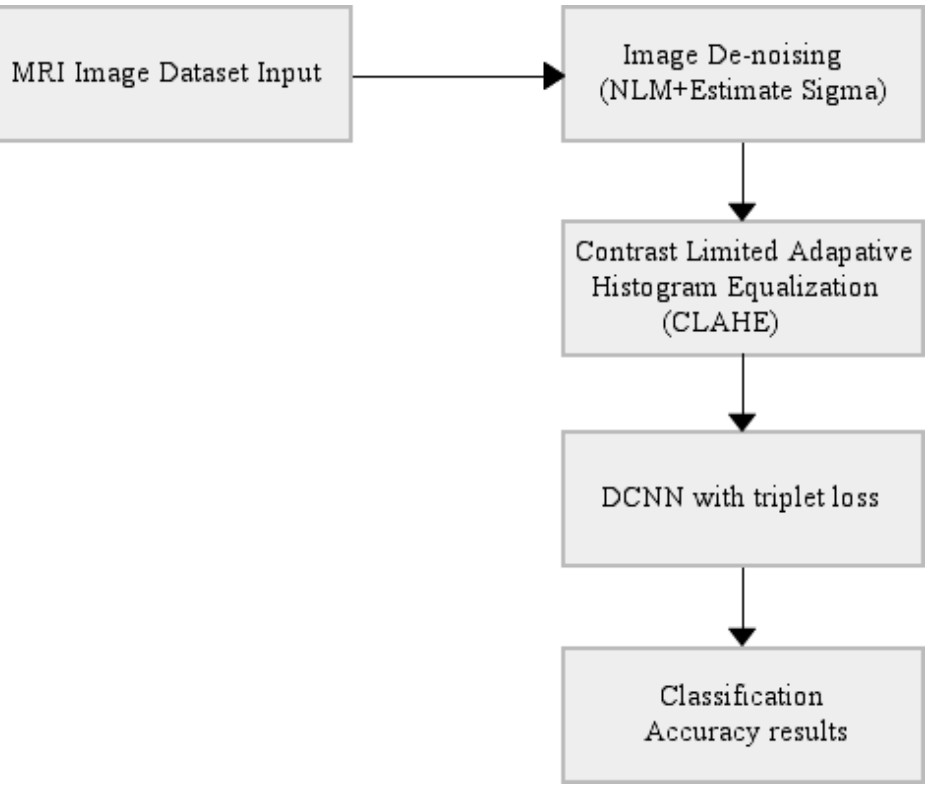

**Figure 3** End-to-end automatically DCNN-based AD classifier block diagram.

The lambda value was chosen as 0.5 because of experimental studies.

## Performance metrics

Accuracy, Sensitivity, and Specificity metrics were used to measure the performance of the proposed deep learning-based classification model. Accuracy is the ratio of correctly predicted samples to the sum of all correct and incorrectly guessed samples; Sensitivity, or recall, is the ratio of positively correctly predicted samples to the sum of negative incorrectly predicted samples and correctly predicted positive samples. Specificity (Spec) represents the ratio of correctly predicted negative samples to the sum of correctly predicted negative and incorrectly predicted positive samples.

## The proposed methodology

The end-to-end automatically AD classifier block diagram is shown in Fig. 3. The section including CLAHE covers the pre-processing stage of the dataset, and the section after CLAHE covers the training and testing of the model.

## The proposed deep learning model

In the proposed deep learning architecture, 2xConvolutional2D blocks with $3 \times 3$ filters are used in all layers before the fully connected layer. In addition, the Xavier

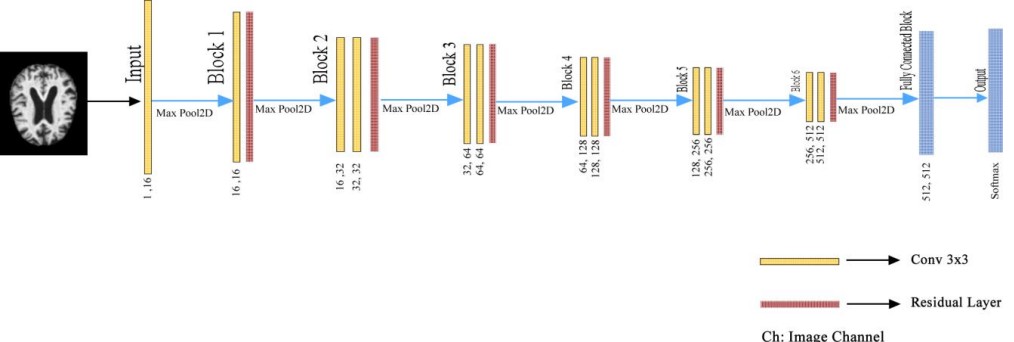

**Figure 4** Detailed representation of the proposed architecture.

was used as the weight initializer function (*Glorot & Bengio, 2010*; *Sharma et al., 2023*; *Nour, Senturk & Polat, 2023*). Group Normalization (GN) is the normalizer, and ReLU is the activation function of the proposed model (*Wu & He, 2020*). *Wu & He (2020)* demonstrated with various experiments how superior the group normalization technique is to other normalization methods.

Besides, ADAM was used as the optimizer to optimize the weights of the proposed model (*Ba & Kingma, 2015*). In the last layer, Softmax was used as an activation function to find out which class the MR image belongs to the AD class. If we pay attention to the proposed model, it is seen that the U-shaped network is formed by taking the deconvolution part and adding the Fully connected network. Deep Neural networks are profound networks due to the multitude of layers. Therefore, the information learned in the first layers can be forgotten. The residual blocks (ResNet Block) ensure that the convolutional layers do not forget the information they learned in the first layers of the network. In the proposed model, residual blocks are followed by $2\times 2$ maximum pooling (max-pool), in which the feature maps' spatial dimensions (height and width) are reduced by half. Max-pool reduces the computational cost by reducing the number of trainable parameters. The layers of the proposed deep learning architecture are shown in Fig. 4. in detail.

## Proposed model's experimental studies

This section explains the experimental studies of the proposed deep learning architecture on the KAGGLE dataset in detail.

### *Employed hardware materials*

Experimental studies of the proposed architecture were conducted with a computer equipped with Intel(R) Core(TM) i5-8300H @ 2.30 GHz CPU, 32 GB RAM, and NVIDIA GTX1050 4 GB GPU. In addition, the deep learning ecosystem consists of artificial intelligence libraries with Python 3.7 programming language based on Anaconda. Libraries and ecosystems are entirely open source. The dataset consists of training, validation, and test datasets. A total of 20% of the training dataset was used as a validation dataset using the

**Table 1  Comparison of parameters and their computational efficiency.**

| Architectures | Parameter (M) |
|---|---|
| VGG16 (*Sharma et al., 2022*) | 134.27 |
| Alexnet (*Loddo, Buttau & Di Ruberto,2022*) | 62.38 |
| Resnet 101 (*Loddo, Buttau & Di Ruberto,2022*) | 44.5 |
| Inception ResNetv2 (*Loddo, Buttau & Di Ruberto,2022*) | 54.5 |
| Proposed Architecture | 5.45 |

**Table 2  Effects of different loss functions on the proposed architecture.**

| Methods | Performance metrics | | |
|---|---|---|---|
| | Accu | Sens | Spec |
| Cross-entropy | 0.966 | 0.966 | 0.982 |
| MAE (*Loddo, Buttau & Di Ruberto,2022*) | 0.932 | 0.932 | 0.934 |
| RMS (*Loddo, Buttau & Di Ruberto,2022*) | 0.956 | 0.959 | 0.957 |
| FL (*Loddo, Buttau & Di Ruberto,2022*) | 0.927 | 0.930 | 0.928 |
| GDL | 0.965 | 0.968 | 0.966 |
| **Our fusion loss** | **0.982** | **0.982** | **0.989** |

scikit-learn library. We performed 5-fold cross-validation to measure the training dataset's heterogeneity and the model's fit.

### Comparison of parameters
The model used in the proposed study and the models used in the other literature were analyzed comparatively regarding the number of parameters. As can be seen from the comparisons in Table 1, the proposed model is the one with the smallest parameter. It can be seen from Table 1 that the model with the smallest parameter is the recommended model.

### Comparison of loss functions
The proposed fusion loss function has been analyzed in comparison with the most used mean absolute error (MAE), root mean squared error (RMSE), cross-entropy, FL, and GDL loss function for classification in the literature. As can be seen from Table 2, the proposed fusion loss function is the most suitable loss function for the proposed architecture.

### Settings and studies on the KAGGLE dataset
Using our deep learning architecture, we performed binary classification, *i.e.,* distinguishing NC and AD, and multi-class classification (NC, very mild AD, mild AD, moderate AD) experimental studies on the KAGGLE dataset. The deep learning network was trained for 15 thousand epochs for these two studies. After 15.000 epochs, the validation accuracy of the network remained constant at around 0.99. The minibatch size is 32, and ADAM is chosen as the optimizer. Numerical information about the data set used in training the proposed model is listed in Table 3. As can be seen from Table 3, there is a significant imbalance between classes. In this case, the benefits of data augmentation methods will be

**Table 3  Slice-based MR image counts used for train, validation, and testing of the KAGGLE data set on which the proposed model is trained and tested.**

| Data set name | NC | vmiD | miD | mD |
|---|---|---|---|---|
| Training data set | 2,050 | 1,075 | 573 | 42 |
| Train+Data Augmented | 7,170 | 4,659 | 2,007 | 146 |
| Validation data set | 510 | 717 | 287 | 10 |
| Test data set | 640 | 448 | 179 | 12 |

limited. Therefore, GDL and FL, which are successful and robust loss functions in class imbalances, are proposed as a solution to the class imbalance of the data set.

## Experimental results and analysis

In this section, experimental results are shared, and comparative analyzes are made with other studies in the literature. In addition, information is given about the deep learning models used for detecting AD and how they train the dataset on the performance criteria they use. A slice-based data set was used in this study. The Kaggle dataset is a low-resolution and slice-based dataset collected on different websites. In addition, class imbalances are also very high in the dataset. Also, the Kaggle dataset's having axial views is another challenge. For these reasons, the Kaggle dataset is very challenging. The most significant difficulty of all the literature and proposed research studies is distinguishing AD from NC in the MCI stage. If AD is successfully detected in the MCI stage, the effects of this neurodegenerative and irreversible disease can be slowed down. In addition, one of the most critical challenges is to detect NCs (very MCI) that may have AD. This study focused on two different experimental studies: (i) distinguishing NC from AD at the MCI stage; (ii) To successfully classifying MR images with NC, vMCI, MCI, and MC. For this reason, articles in the literature that made these two different experimental studies and analyzed their results were examined. Validation accuracy and validation loss values obtained using the categorical generalized focal membrane loss function of the proposed model are shown graphically in Figs. 5 and 6.

The comparative performance results of our proposed fusion loss-based deep learning architecture are shown in Tables 4 and 5. The proposed deep learning architecture achieved accuracy values of 0.973 in binary classification and 0.982 in multi-class classification. At the same time, it obtained high sensitivity and specificity values compared to other studies in the literature.

## DISCUSSION

This study proposes a deep learning-based model with fusion loss for diagnosing and classifying AD. The Kaggle dataset, the most challenging AD dataset, was used to compare the proposed architecture with other studies in the literature. High accuracy values of up to 0.99 were obtained in other publicly available datasets (OASIS and ADNI) used in the literature. The Kaggle dataset has four classes: vmiD, miD, mD, and NC. The proposed fusion loss deep learning model showed a higher performance than other studies in the literature. In addition, Ensemble-based deep learning algorithms are generally emphasized

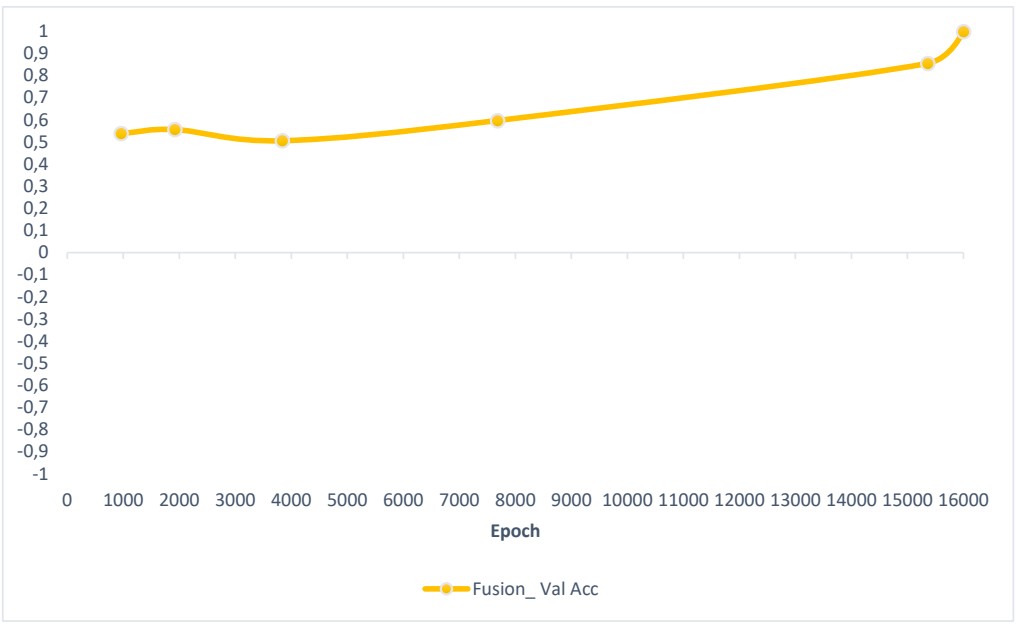

**Figure 5** ResBlock fully connected CNN validation loss and validation accuracy results.

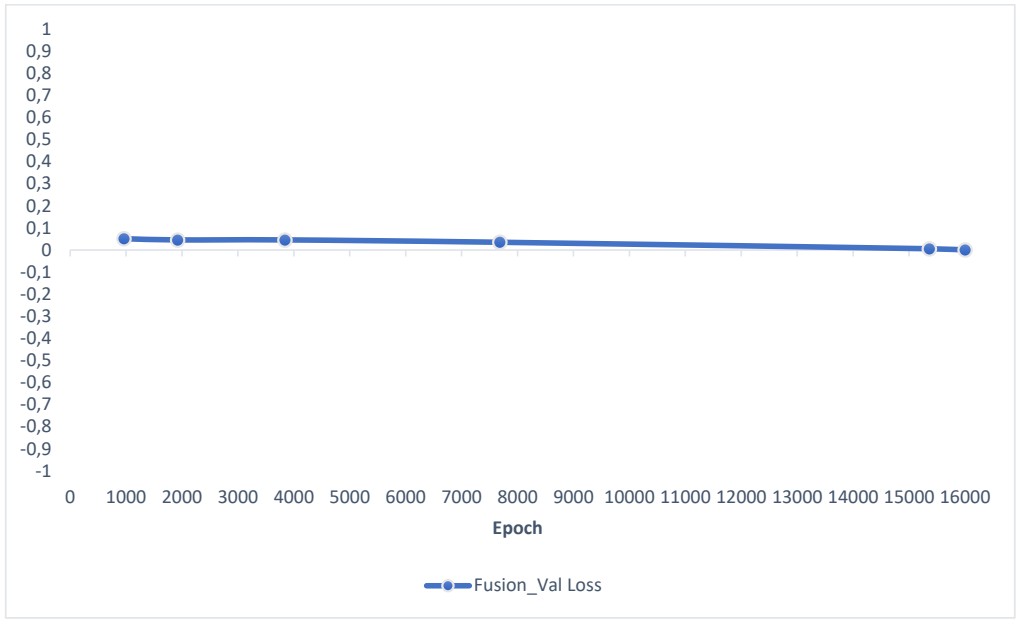

**Figure 6** ResBlock fully connected CNN validation loss and validation accuracy results.

**Table 4  Experimental results for NC and AD binary classification.**

| Methods | Performance metrics | | |
|---|---|---|---|
| | **Accu** | **Sens** | **Spec** |
| Deep-Ensemble (*Loddo, Buttau & Di Ruberto,2022*) | 0.966 | 0.966 | 0.982 |
| AlexNet (*Loddo, Buttau & Di Ruberto,2022*) | 0.897 | 0.898 | 0.897 |
| ResNet-101 (*Loddo, Buttau & Di Ruberto,2022*) | 0.961 | 0.961 | 0.961 |
| Inception-ResNet-v2 (*Loddo, Buttau & Di Ruberto,2022*) | 0.912 | 0.914 | 0.912 |
| **(Our Methodology)** | **0.973** | **0.975** | **0.988** |

**Table 5  Experimental results for multi-class classification(NC, vmiD, miD, mD).**

| Methods | Performance metrics | | |
|---|---|---|---|
| | **Accu** | **Sens** | **Spec** |
| Deep-Ensemble (*Loddo, Buttau & Di Ruberto,2022*) | 0.971 | 0.967 | 0.982 |
| Neural Nets with VGG16 (*Sharma et al., 2022*) | 0.904 | 0.905 | 0.904 |
| AlexNet (*Loddo, Buttau & Di Ruberto,2022*) | 0.893 | 0.906 | 0.817 |
| ResNet-101 (*Loddo, Buttau & Di Ruberto,2022*) | 0.965 | 0.978 | 0.961 |
| Inception-ResNet-v2 (*Loddo, Buttau & Di Ruberto,2022*) | 0.897 | 0.901 | 0.856 |
| **(Our Methodology)** | **0.982** | **0.982** | **0.989** |

in the literature. The deep learning architecture proposed in this paper also dramatically reduces computation. However, the minibatch could not be increased enough due to hardware limitations. In addition, resizing was not done because the resolution of the data line was low. Therefore, the number of epochs determined for training the proposed model may be higher than those used by other models. However, the parameter number of the proposed architecture is considerably smaller than the parameter numbers of different architectures in the literature.

## CONCLUSIONS

This study proposes a fusion loss deep learning model using the group normalization technique with residual blocks to detect three different AD disease classes (vmiD, miD, mD). The proposed model is trained on the KAGGLE dataset, a very challenging dataset that includes three other dementia classes (vmiD, miD, mD) and normal cognitive status. We tried to solve the large class imbalance in the Kaggle dataset using the Categorical Generalised Focal Dice Loss function proposed in this study. In the proposed model, FL and GDL loss functions are used successfully in multi-class segmentation by fusing them. FL and GDL were used separately in the proposed model, and 84% to 88% accuracy was achieved, while an accuracy value of 0.982 was obtained in the model with the fused loss function. Compared to the latest technology studies in the literature, the proposed model has achieved very high success in binary classification (NC and AD) and multi-class classification (vmiD, miD, mD, NC). Although the proposed architecture showed high performance on a challenging dataset, it must be tested on an MR device in real-time.

### Funding

This work was supported by the Deputyship for Research & Innovation, Ministry of Education in Saudi Arabia through project number (IF-PSAU-2021/01/18563). The funders had no role in study design, data collection and analysis, decision to publish, or preparation of the manuscript.

### Grant Disclosures

The following grant information was disclosed by the authors:
The Deputyship for Research & Innovation, Ministry of Education in Saudi Arabia: IF-PSAU-2021/01/18563.

### Competing Interests

The authors declare there are no competing interests.

### Author Contributions

- Adi Alhudhaif conceived and designed the experiments, performed the experiments, analyzed the data, performed the computation work, prepared figures and/or tables, authored or reviewed drafts of the article, and approved the final draft.
- Kemal Polat conceived and designed the experiments, performed the experiments, analyzed the data, performed the computation work, prepared figures and/or tables, authored or reviewed drafts of the article, and approved the final draft.

### Data Availability

The Alzheimer's Dataset is available at Kaggle: https://www.kaggle.com/datasets/tourist55/alzheimers-dataset-4-class-of-images.

### Supplemental Information

Supplemental information for this article can be found online at http://dx.doi.org/10.7717/peerj-cs.1599#supplemental-information.

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
