# Peer review of "Residual block fully connected DCNN with categorical generalized focal dice loss and its application to Alzheimer’s disease severity detection"

_PeerJ Computer Science, doi:10.7717/peerj-cs.1599_

## Round 0.1 · original submission · Major Revisions

Dear authors

Please improve the quality of the paper as suggested by the experts. Please also elaborate on the methodology in more detail, highlighting the novelty of the work.

Please also improve the technical language of the manuscript.
Thanks

Reviewer 1 ·

Basic reporting

In this paper, the authors proposed deep learning model supported by a fusion loss model that includes fully connected layers and residual blocks to solve the above-mentioned challenges. The proposed model has been trained and tested on the publicly available T1-weighted MRI-based KAGGLE dataset. The proposed model is interesting, and then the obtained results are good. But there are some problems with the paper:

a) Why did you use the proposed method? Please give some examples explaining the working of the proposed model.

b) Please give a block scheme of the used model in the paper. This structure should be given as an algorithmic.

c) The NN model should be explained in the paper. Please give the parameters in the paper.

d) Future works should be given in the conclusions.

Experimental design

1. In section III, It is suggested that the authors describe the parameters in bullets.
2. It is suggested that the authors should elaborate on the results in more detail.

Validity of the findings

The experimental Setup is Good.

Additional comments

- Already given in Basic reporting

Reviewer 2 ·

Basic reporting

The author proposed a novel hybrid deep learning model.

The proposed method seems very good.

Experimental design

In this study, the authors have conducted many experiments.

Validity of the findings

The obtained findings are very good.

Additional comments

The authors offer an automated artificial intelligence method for their problem. The presented study is very important, considering that other methods in the literature are generally processed manually. The overall narrative is good, but some minor corrections are needed.

1- The English of this paper should be polished.

2- How were the features that can be both output and input selected?

3- Please add training parameters.

4- please expand discussions.

Please add some information about the results of your approach in the abstract section.

Please add more discussion.

Please give an example showing the proposed system work.

Reviewer 3 ·

Basic reporting

The paper is written in a clear English language without ambiguity. The format of the references should be revised because it is inconsistent.

Experimental design

In this paper, the authors have conducted many experiments. To classify MRI images, they proposed a different systemdesign based on combining deep learning models and MRI images.

Validity of the findings

The findings are very excellent. To validate the experimental results, they used the performance metrics.

Additional comments

1) Indicators display must be consistent. Please note that the F function sequence must be in the same order throughout the article.
2) Why is used for deep learning classifier? Please explain it in detail in the paper.
3) How were the features that can be both output and input selected?
4) Please add training parameters.
5) please expand discussions.
6) Furthermore, where are the limitations of your study?

---

## Round 0.2 · accepted · Accept

Congratulations, your article is recommended by the experts.


Reviewer 1 ·

Basic reporting

The authors made all the issues in the revised paper. It looks great.

Experimental design

The authors made all the issues in the revised paper. It looks great.

Validity of the findings

The authors made all the issues in the revised paper. It looks great.

Additional comments

The authors made all the issues in the revised paper. It looks great.

Reviewer 2 ·

Basic reporting

The authors have addressed the issues in the revised paper.

Experimental design

The authors have addressed the issues in the revised paper.

Validity of the findings

The authors have addressed the issues in the revised paper. It is done.

Additional comments

The authors have addressed the issues in the revised paper.

Reviewer 3 ·

Basic reporting

The authors conducted all the comments that arose in the revised paper. The revised paper looks great.

Experimental design

It is done.

Validity of the findings

It is done.

Additional comments

The revised paper is good. It could be accepted.